# Synthesis Methods of Obtaining Materials for Hydrogen Sensors

**DOI:** 10.3390/s21175758

**Published:** 2021-08-26

**Authors:** Izabela Constantinoiu, Cristian Viespe

**Affiliations:** 1Laser Department, National Institute for Laser, Plasma and Radiation Physics, Atomistilor 409, RO-077125 Magurele, Romania; izabela.constantinoiu@inflpr.ro; 2Faculty of Applied Chemistry and Materials Science, University Politehnica of Bucharest, RO-011061 Bucharest, Romania

**Keywords:** surface acoustic wave, sensors, hydrogen, sensor, sol-gel, co-precipitation, spin-coating, pulsed laser deposition

## Abstract

The development of hydrogen sensors has acquired a great interest from researchers for safety in fields such as chemical industry, metallurgy, pharmaceutics or power generation, as well as due to hydrogen’s introduction as fuel in vehicles. Several types of sensors have been developed for hydrogen detection, including resistive, surface acoustic wave, optical or conductometric sensors. The properties of the material of the sensitive area of the sensor are of great importance for establishing its performance. Besides the nature of the material, an important role for its final properties is played by the synthesis method used and the parameters used during the synthesis. The present paper highlights recent results in the field of hydrogen detection, obtained using four of the well-known synthesis and deposition methods: sol-gel, co-precipitation, spin-coating and pulsed laser deposition (PLD). Sensors with very good results have been achieved by these methods, which gives an encouraging perspective for their use in obtaining commercial hydrogen sensors and their application in common areas for society.

## 1. Introduction

Hydrogen is a current topic for the world of science. Due to its energy potential, it is becoming the source of a new generation of fuel. It is a renewable, clean, environmentally friendly and highly efficient source of energy [1,2,3,4]. Many current environmental problems are intended to be reduced by replacing fossil fuels with energy generated by hydrogen [2,3,4,5,6]. Until now, hydrogen has been used as a rocket fuel in aerospace programs [4,7]. BMW, Honda, Toyota and Hyundai have already introduced hydrogen-powered vehicles based on hydrogen fuel cells on the market [4,8]. Other industries in which hydrogen is used because of its reducing properties are the chemical industry, aerospace engineering, pharmaceutics, metal reduction, petroleum extraction and in synthetic fuels [8,9,10,11,12] (Figure 1). Thus, its increasingly frequent utilization in various areas leads to the need to ensure safety during use.

Up to a concentration of 4%, hydrogen has no risk of flammability or explosion. Above this value, the risk of inflammation increases, and at even higher concentrations, it becomes explosive [9,13,14,15]. It must also be taken into account that hydrogen has a very small molecule with a high coefficient of diffusion in air (0.6 cm^2^/s at 0 °C), being thus capable of permeating many materials [8]. This aspect leads to the possibility of uncontrolled accumulations in certain spaces, which favors the production of explosions. In addition, high hydrogen concentrations lead to a decrease in the amount of oxygen and consequently to suffocation, neuronal apoptosis and other neuronal disorders [2,16,17]. Because hydrogen cannot be detected directly by the human senses by odor or color, and due to its small atomic radius [18,19], it is necessary to perform this detection by means of specialized sensors [3,4,19,20]. With regard to the lack of color and odor of hydrogen, current studies aim to attribute these characteristics to it [21]. This is especially applicable for certain areas where its release and accumulation are imminent, in order to facilitate its detection, including by human means.

The performance criteria of a hydrogen sensor are as follows: detection in the range of concentrations 0.01–10% for safety and 1–100% for fuel cells, selectivity to other reducing gases such as NO, CO, H_2_S, etc., high sensibility, high accuracy, short response and recovery times, suitable operating temperature, (preferably room temperature operation) stability to environmental factors (such as temperature and humidity), repeatability, long-term stability and low cost [4,22]. Some of the types of sensors which meet these criteria are studied in the literature for hydrogen detection: surface acoustic wave sensors [23,24,25], resistive [26,27,28], conductometric [29,30,31], optical [32,33,34] or catalytic sensors [35,36,37]. Each of these sensors has different operating principles, but they are similar in that each of them uses a sensitive material to identify the presence of the analyte. Consequently, the sensitive material becomes an important performance factor of the sensors, and its development occupies an important place in the interests of the researchers. Some aspects related to the sensitive material of the sensors will be discussed in the paper. There are some generally valid characteristics that must be taken into account in the design of a material in order to obtain the best possible results: composition, structure, microstructure and morphology. These characteristics are influenced by the synthesis method and the conditions chosen for the synthesis process. The categories of materials most used for hydrogen sensors are semiconductor metal oxides [38,39], metals [40,41,42], polymer [43,44] and composite materials [27,45,46,47]. Each of these materials has advantages and disadvantages that can vary depending on the type of sensor used for detection and on the environmental conditions in which the sensors are tested. Another important aspect is the fact that the synthesis methods are specific for each category of material. Among the most used synthesis methods are sol-gel [48,49], evaporation [1], RF magnetron sputtering [50], DC magnetron sputtering [51], precipitation [52], electrospinning [53], pulsed laser deposition (PLD) [54], thermal oxidation [55], hydrothermal [28] and in situ self-assembling [56]. Table 1 presents some results of hydrogen resistive sensors obtained using these methods. It can be observed that the performance of the sensors is influenced both by the chosen synthesis method and by the composition of the material.

The purpose of this paper is to summarize the latest studies on the development of new materials by means of simple and relatively accessible methods, which encourages the development of hydrogen sensors. The methods explored here are sol-gel, co-precipitation, spin coating and PLD.

## 2. Synthesis Methods

The development of materials with new properties that bring performance in today’s technology is a major concern for research. The synthesis of new materials or materials with new properties automatically leads to the improvement of synthesis methods. Through unconventional synthesis methods, new or improved properties of the materials have been obtained, which then led to the progress of technology. The main characteristics of materials obtained by such methods and which have led to advancements in the field, are small particle sizes (below 100 nm), different types of morphology, the ability to control doping concentrations or complex compositions, obtaining materials with several phases or the synthesis of 2D materials [57,58,59].

In the domain of sensors, especially for hydrogen sensors, several methods of synthesis and deposition have been used most often: sol-gel [60,61], hydrothermal [62], thermal evaporation [63,64], PLD [65,66], magnetron sputtering [67,68], co-precipitation [69,70] and spin coating [71,72].

Given that hydrogen is a very small molecule with a high degree of permeability through various materials and that its selective detection is difficult to achieve, the synthesis of sensitive material for sensors with improved performance is still a challenge [73,74,75].

In order to be able to correctly choose a synthesis method for a material, it is necessary to know the processes involved and the major factors of influence on the final properties of the materials. In this way, the synthesis parameters can be set so as to achieve the characteristics necessary for a particular application.

Four methods of synthesis of materials and thin films for hydrogen sensors will be discussed in this work: sol-gel, co-precipitation, spin coating and PLD. These methods were chosen because they are easily accessible, and they have already achieved promising results in the field of sensors.

### 2.1. Sol-Gel

The sol-gel synthesis method is an unconventional synthesis route, known for the possibility of controlling the final characteristics of materials. It allows the control both from the compositional and morphological point of view by varying the synthesis conditions [76,77,78].

The sol-gel method is based on two types of reactions: the hydrolysis reaction, followed by the polycondensation reaction [78,79]. Practically, a solution is transformed into a gel, then, after a thermal process, it reaches the powder stage (Figure 2) [78,79]. The hydrolysis reaction starts from an alkoxide dissolved in a solvent. It continues with polycondensation, which take place after the addition of a small amount of water and leads to the formation of a polymer chain and therefore of the gel-type material. Following this, a thermal process, most often in stages, removes the liquid phase from the gel and a powder is obtained [78,79,80]. There are several processing options after obtaining the sol, depending on the type of deposition to be made. Thus, the sol can be deposited on the substrate, forming a xerogel, which after a heat treatment becomes a dense film [64]. Another way of processing the sol, as it can be seen in Figure 2, is the one characteristic of this method: gelation, evaporation (obtaining a xerogel or an aerogel), followed by a heat treatment at high temperatures to obtain a material as dense as possible. There are also processing methods that skip the gel phase, such as precipitating the sol or making fibers by electrospinning [78,80,81].

Figure 3 shows the advantages and disadvantages of the sol-gel synthesis method [70,80,82]. Whatever the sol-gel synthesis option chosen, among the advantages of this method are: control of the reactions involved in the synthesis and obtaining materials with homogeneity, even in systems with a large number of components [83,84]. Another very important advantage of this method is that it does not involve special conditions for synthesis, such as ensuring a certain pressure or a certain type of atmosphere [70,80,82].

Han et al. [14] developed Pd-WO_3_ multilayer composite films by the sol-gel method. Using Pluronic F127 as template, the porosity of the films increased, offering a larger specific surface area, favorable for gas detection. The presence of Pd led to a sensitivity of 346.5 higher than in the case of only WO_3_. This property was improved also by the formation of p-n heterojunctions.

Yadav et al. [20] obtained ZnO films by combining sol-gel and spin coating methods, under different synthesis conditions. They analyzed the different morphologies obtained due to the variation of the synthesis conditions and they found that the morphology of the sensitive material of hydrogen sensors has a great influence on their performance. Abdullah et al. [70] analyzed the influence of a sol-gel SnO_2_ film on the sensitivity of a thin layer Ga_2_O_3_ hydrogen sensor. They obtained remarkable results at room temperature, and the tests at different temperatures indicated a significant improvement in sensor results, which confirms that the use of the sol-gel synthesis method is a facilitator for the improvement of hydrogen sensors. As evident from the responses of sensors given in Figure 4a, there is a large difference between the performance of the sensors at various temperatures and their responses at room temperature. Since one of the targets in the development of sensors is that they are capable of operating at room temperature, it is very important that the sensors respond at room temperature. In terms of response and recovery times (Figure 4b), the sensor tested at room temperature recorded values not far from the values obtained at high temperatures.

The sol-gel method is also recommended for the synthesis of materials with complex composition, because it offers homogeneity and good control over the composition of the material [85,86]. Kostadinova et al. [82] synthesized a material in the 85SiO_2_-9P_2_O_5_-6TiO_2_ oxide system by the sol-gel method. They obtained Si_5_P_6_O_25_ as the predominant phase and SiP_2_O_7_ as secondary phase of the material. In addition, through the SEM images it was possible to observe a porous morphology. Thus, by joining such molecular structures with the porous morphology, the conditions for the penetration of hydrogen molecules are improved.

One of the main factors influencing the performance of the sensors is the large specific surface area of the sensitive material of the sensor. In order to obtain such morphologies, difficult, complicated and expensive methods are usually involved. Mane et al. [87] obtained ZnO flower-like nanostructures with large porosity using the sol-gel drop-casting method. Their results indicated considerable improvements due to this type of morphology. Taking into account the simplicity and low costs of the method, the perspective of using this method on a large scale for hydrogen sensors becomes closer to realization.

As shown, the sol-gel synthesis method is a very promising alternative for developing materials with new compositions and suitable morphologies to achieve high-performance hydrogen sensors, including at room temperature. In addition, this method is adaptable enough to be used for several types of sensors.

### 2.2. Co-Precipitation

Co-precipitation is one of the most common and simplest methods used to obtain nanoparticles. It is a method that does not require advanced techniques, high energy consumption or ensuring special or difficult conditions [88,89].

The method involves the precipitation of metal cations in the form of hydroxides, carbonates, citrates or oxalates. The final powder is obtained following a heat treatment of calcination [90,91]. The conditions of the heat treatment depend on the type of the material synthesized. However, due to the fact that the heat treatment of calcination does not generally require very high temperatures, the formation of particles with small dimensions is favored [91].

During the synthesis by co-precipitation, the following processes take place simultaneously: nucleation, growth, coarsening and agglomeration. Because during nucleation very small particles are formed, this is an important step of the synthesis process. The next steps of growth, coarsening and agglomeration represent stages that influence the size, morphology and some of the final properties of the material [92,93].

Practically, as can be seen in Figure 5, this process starts from the precursors containing the cations which are intended to be in the final powder, which are homogenized in a suitable solvent. When the solution reaches supersaturation the nucleation process begins, followed by the growth mechanism. The precipitate obtained is then filtered and subject to the heat treatment of calcination, obtaining the final powder of the material. Synthesis products are insoluble species under supersaturation conditions.

The final characteristics of the nanoparticles are strongly influenced by the concentration of the reactants, the time and the order in which the reactants are added to the solution, the calcination temperature, the pH of the solution and the use of the surfactants. In addition, it is important that the solubility values of the reactants should be compatible for the synthesis to work and in order to obtain the established material. There are other external factors that can influence the quality of the powder obtained: stirring speed and vibration, exposure to light and cleanliness of glassware [90,94].

Figure 6 mentions the advantages and disadvantages of the co-precipitation synthesis method. Two of the major disadvantages of the co-precipitation synthesis method are the impurities that are a possible secondary result of the reactions, and the difficulty of controlling the rate of nucleation and particle growth, resulting in a wide distribution of particle sizes [92,95]. A major advantage of the co-precipitation synthesis method is that it is a simple method, and one through which a wide variety of nanocomposite materials can be obtained directly. For applications in the field of sensors, co-precipitation is an accessible method which leads to a low price of production.

Fomekong et al. [68] studied the influence of TiO_2_ doping with different concentrations of Ni, by the method of co-precipitation. The 0.5% Ni doped sensor indicated the best response in terms of sensitivity for hydrogen, as well as for selectivity.

Sharma et al. [95] exploited the potential of halloysite nanotubes (HNT) by doping with Fe_3_O_4_ and Pd. The synthesis process took place in two stages. First, by the reduction-precipitation method, HNTs were enriched with Fe_3_O_4_ which has the role of increasing the sensitivity of the sensor. Pd doping was performed by the hydrothermal method and aimed at providing selectivity to the sensor. Through the SEM images made after each stage of Fe_3_O_4_-HNT-Pd synthesis (Figure 7), the changes in morphology can be observed, as well as the uniformity of the deposits.

By the co-precipitation method, Pathania et al. [69] synthesized a material with complex composition: Ni_0.5_Zn_0.5_W_x_Fe_2−x_O_4_ (x = 0.0, 0.2, 0.4, 0.6, 0.8, 1.0). From SEM images, the formation of a morphology with well-defined granules was identified, which offers a large specific surface area to the material; EDX spectra confirmed the presence of all elements introduced in the system. The sensors that had tungsten in their composition indicated selectivity for hydrogen.

Fomekong and Saruhan [96] reported a co-precipitation synthesis of Co- doped TiO_2_ materials in different percentages. Hydrogen tests indicated a high sensitivity at 200 ppm and 600 °C compared to NO_2_. Further, undoped TiO_2_ indicated the best results under the mentioned conditions. This indicates that the synthesis method favored the obtaining of a TiO_2_ material with improved properties for hydrogen detection.

Low energy consumption during the synthesis of a material is an advantage for the synthesis method, and Shaposhnik et al. [97] synthesized by co-precipitation SnO_2_ and TiO_2_ at low temperatures. They obtained structural homogeneity of materials and remarkable results of sensor tests with this material at concentrations between 1–500 ppm hydrogen.

The presented works indicate that co-precipitation is a method that allows the obtaining of materials with complex compositions, doping different materials with elements that lead to obtaining the selectivity of sensors for hydrogen.

### 2.3. Spin Coating

The modification of surfaces by the deposition of thin films or their functionalization has become a research direction of great interest, thus contributing to a great extent to the development of fields such as medicine, sensors and different areas of technology. One of the ways to realize this modification is by applying thin films to the surface of different materials [98,99].

Among the most commonly used methods for synthesizing thin films is spin coating. It is a method that allows the production of uniform thin films, with thicknesses of the order of micrometers and nanometers, on flat-shaped substrates [100,101]. It allows the deposition of both organic and inorganic materials [102,103]. This method uses centrifugal force to evenly spread the material solution over the entire surface of the substrate [104].

A typical spin coating deposition process (Figure 8) begins with the preparation of the deposition material in a slightly volatile solvent, which needs to have a certain viscosity to allow uniform deposition on the entire substrate. The substrate is placed in the spin coater on a chuck that rotates the sample and it is fixed by suction. A certain amount of the material solution is dropped onto the center of the substrate, which is then accelerated (up to 8000 rpm), following a certain program. The material is spread by centrifugal force on the surface of the substrate, and after evaporation of the solvent, the thin film is obtained. For multilayer structures this process can be repeated [104,105].

Two important aspects that lead to the formation of a thin film are the viscous force and the surface tension [100]. The final properties of the films are also strongly influenced by other characteristics, such as: material solution properties, substrate properties, spin speed and acceleration [106]. The thickness of the film is one of the properties that can be controlled in this process, depending on the few parameters that influence it. Equation (1) [100] shows how the film thickness is influenced by other parameters and can be controlled, where *h*—thickness, $\rho_{A}$—density of volatile liquid, *η*—viscosity of solution, *m*—rate of evaporation and *ω*—angular speed.
(1)$$h=\left(1-\frac{\rho_{A}}{\rho_{A0}}\right)\cdot {\left(\frac{3\eta \cdot m}{2\rho_{A0}\omega^{2}}\right)}^{1/3}$$

A simpler equation (Equation (2)) [100] can be used to calculate the film thickness, taking into account that the evaporation rate can be determined experimentally, where *B* is a constant and experimentally calculated parameter with values in the range of 0.4 to 0.7. This equation obviously shows that the higher the rotation speed, the smaller the film thickness [87].
(2)$$h=A\omega^{-B}$$

Figure 9 highlights the advantages and disadvantages of this deposition method. It is worth mentioning as an important advantage that it is a simple and inexpensive method, which leads to obtaining thin and uniform films. An important disadvantage is given by the low efficiency of the amount of material deposited. About 95–98% of the material is flung off of the substrate during spin process, with only about 2–5% of the amount of material remaining on the substrate [104,107].

Bai et al. [108] chose to combine the synthesis of SnO_2_ nanospheres modified with Sn_2_O_3_ by the solvothermal method with the spin coating deposition method. Thus, for hydrogen detection, the mentioned material was deposited by spin coating on the surface of a resistive sensor, which then indicated the presence of hydrogen down to a concentration of 100 ppm.

He et al. [109] have developed a strategy for the composite Pd (II)@alkyne-PVA/d-Ti_3_C_2_T_x_, which is a material with promising properties for hydrogen detection. This material was applied on the sensor substrate by spin coating, obtaining a uniform sensitive film both from a compositional point of view and in terms of surface properties. The SEM images, element mapping and EDX spectrum in Figure 10 confirm these aspects.

Choi et al. [110] have improved the properties of a hydrogen sensor using a ZnO nanoparticles layer, deposited by spin coating. The cross-sectional TEM image (Figure 11) shows the uniformity of the layer, and a thickness of the ZnO layer deposited by spin coating of approximately 170 nm.

In Jung et al. [111], the spin method is used to make a thin Pt-CNT (carbon nanotube) composite film for a hydrogen sensor. Pt-CNT films with a thickness of 6 nm obtained the best results in the detection of hydrogen at room temperature, at concentrations of 3–33%.

Inpaeng et al. [112] developed hydrogen sensors based on a dispersion of Pd nanoparticles on graphene sheets, by spin coating methods. The sensors produced by this method were tested at several concentrations of hydrogen at room temperature, with a detection limit down to 1 ppm.

In view of these results, it can be stated that the spin coating deposition method is a simple method by which thin films of very good quality can be made in a short time and with low costs, including for sensor applications in the detection of hydrogen.

### 2.4. Pulsed Laser Deposition (PLD)

PLD is a technique known especially for obtaining high-quality thin films. Over time, this technique has been developed to obtain different types of nanostructures. Taking into account the properties offered by nanostructured materials and the development of technology, most often with miniaturized devices, this technique of synthesis and deposition of thin films remains one of the options of specialists.

Because it is a synthesis method that respects a series of necessary properties in fields that require purity, stoichiometry and a good control of the morphology, synthesis by PLD is used in fields such as biomedical applications or sensors [113,114,115].

The principle of operation of a PLD installation is described in Figure 12. A high-energy laser beam irradiates the surface of the target material, where the laser energy is transferred as electronic excitation, and subsequently transformed into thermal, chemical and mechanical energy, which leads to ablation. As a result of ablation, a plasma is formed with species from the target material (electrons, ions, atoms and molecules). It reaches the substrate positioned in front of the target; the substrate has the possibility to be heated and/or rotated, thus favoring the growth of the film and covering of the entire substrate [116,117,118,119].

It is a thin film synthesis method that allows the variation of several parameters, thus influencing the final properties of the film. There are some important features that must be taken into account to achieve good quality films, regardless of the final particular properties of the films: these refer to the capacity of the material to absorb the laser beam, the plasma dynamics and the deposition of the ablated material on the substrate, the nucleation and the growth of the film [117,118,120].

Regarding the obtaining of films with particular characteristics for certain applications, there are several parameters that can be varied. Some of these are laser parameters: laser energy, fluence, pulse duration and wavelength. The rest of the parameters that can be varied are related to the deposition installation: the deposition chamber pressure, the distance between the target and the substrate, the gas in which the deposition is realized [121,122]. These parameters that can be varied represent a very big advantage for this method of synthesis of thin films, because it makes it applicable in many fields and allows a wide study for optimization.

According to the table in Figure 13, there are a number of advantages that make PLD a frequently chosen method for the synthesis of thin films with special properties, such as very good uniformity, purity and control of stoichiometry and morphology. The slightly weaker points of this method must also be taken into account: it is a method that requires high energy consumption, the deposited surfaces are relatively small and often droplets are obtained, which are a disadvantage for the field of sensors [123,124,125].

A synthesis flow for the PLD method can be considered (Figure 14) that starts from the synthesis of the target material by different methods. The obtained target is located inside the PLD deposition chamber, which ensures the atmosphere required for the deposition: the gas in which the deposition is made and its pressure. After the substrate is placed at a certain distance from the target, the parameters of the laser are fixed. During deposition, the position of the laser beam on the target is continuously changed in order to avoid target erosion, and deposition of the thin film onto the substrate is obtained.

This synthesis method has been widely used for applications in the field of sensors, including hydrogen sensors. The possibility to control the morphology of the films proved important for Constantinoiu et al. [122] for the synthesis of sensitive films for surface acoustic wave sensors. Hydrogen sensors with sensitive TiO_2_ and Pd films have been developed. In order to optimize their morphology, depositions were made at several pressures, both for TiO_2_ (Figure 15) and for Pd. For both types of materials, the influence of pressure on the morphology of the materials was clearly observed. Figure 15 shows the evolution of the TiO_2_ film morphology from the pressure of 100 mTorr O_2_ up to 600 mTorr. Because the porous morphology favors the penetration of the gas molecules in the film volume, and thus the sensitivity of the sensor increases, the authors in [118] established that the TiO_2_ film deposited at 600 mTorr O_2_ was the optimal one (Figure 15e,f). Pd multilayer sensors were made starting from TiO_2_ deposited at 600 mTorr O_2_ together with Pd deposited under several pressure conditions. The results of tests of surface acoustic wave sensors in the presence of various hydrogen concentrations indicated that as the porosity of the films increases, the sensors are more sensitive. The PLD deposition method thus presents a valuable advantage for the sensor field: that of controlling the morphology of the films.

In addition to the pressure of the gas inside the deposition chamber, another important factor in determining the morphology of the films is the heating of the substrate. This influences the growth of the film, in terms of its morphology and structure, and therefore the properties of the sensors [24]. 

Nishijima et al. [126] developed optical sensors with Pt-WO_3_ sensitive material through PLD, in different proportions of Pt and WO_3_. In addition to the importance of the presence of the Pt catalyst in increasing the sensitivity to hydrogen, it was found that the sensors recorded a response for hydrogen at a concentration of 10 ppm, with a response time of 20 s. The importance of a catalyst for hydrogen detection was also confirmed by Koga [127], who improved the mesoporous Co_3_O_4_ surface of a resistive sensor with Pd. The influence of Pd synthesized by laser ablation was studied through a series of tests in which the size and amount of Pd was controlled.

PLD is a method that offers the important advantage of controlling the morphology of deposited films by varying the synthesis parameters. It is also very important to note that multilayer films can be obtained, which maintain the stoichiometry of the target, while also providing purity. For the field of sensors, PLD is a method that can offer new perspectives for obtaining sensors with outstanding performance.

## 3. Conclusions

Hydrogen is one of the most environmentally friendly energy sources and is already applied in various fields, such as transportation, power generation, metallurgy, etc. However, hydrogen has a risk of inflammation and explosion at a concentration of more than 4% in a closed environment. Therefore, control of the concentration of hydrogen in a given environment is very important, and the development of sensors with high sensitivity and selectivity is currently a goal of great interest.

The first step in the development of such sensors starts from the development of the materials used for this purpose and their properties. Using the sol-gel and co-precipitation methods of synthesis, nanomaterials are obtained which make possible the detection of hydrogen at the lowest possible concentrations due to their large specific surface. The deposition of such advanced nanomaterials onto the substrates of sensors by methods such as spin coating or PLD ensures the formation of a layer or multilayer with uniformity and control over the thickness.

The synthesis of materials by the sol-gel method ensures the obtaining of some materials with complex composition, but also with controlled morphology, due to the applied heat treatment. The precursors used in the sol-gel method generally have high costs. Instead, those used for co-precipitation synthesis have low prices, which is an advantage for this synthesis method. Additionally, the co-precipitation is distinguished by the simplicity of the method that still allows the obtaining of doped materials or those with complex compositions.

The synthesis of thin films on the surface of the sensor devices can be achieved by the simple spin coating method. As shown, it allows the synthesis of high-quality films, requiring low costs. PLD deposition, although a method that involves higher energy consumption, allows a better control of the film morphology during the deposition, by the possibility of varying a series of parameters: the gas pressure, wavelength or frequency of the laser, target-substrate distance and temperature of the substrate to be deposited.

These methods, being relatively inexpensive and accessible, encourage the development of hydrogen sensors, which leads to the implementation of hydrogen use in many areas of activity in society.

## Figures and Tables

**Figure 1 sensors-21-05758-f001:**
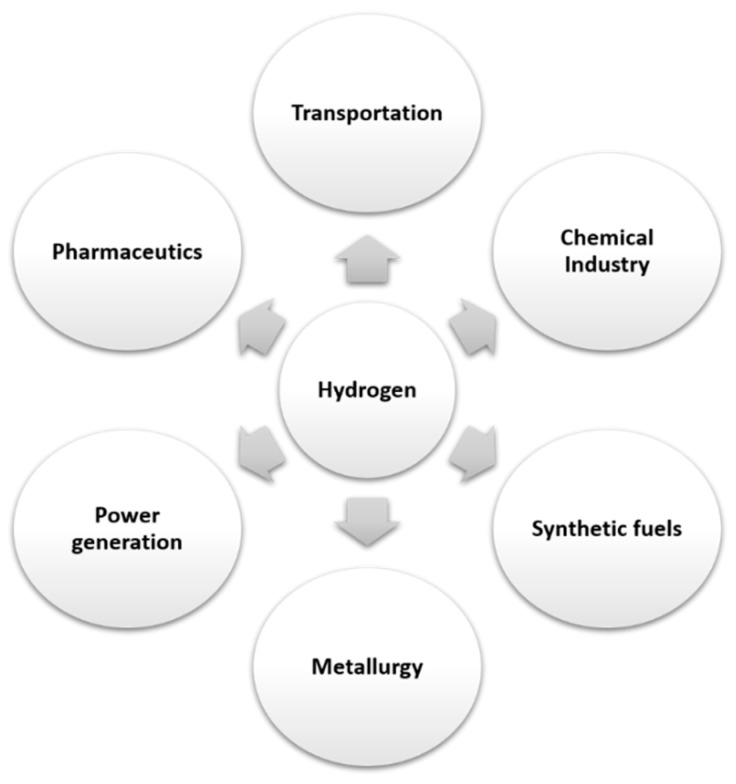
Areas of applicability of hydrogen.

**Figure 2 sensors-21-05758-f002:**
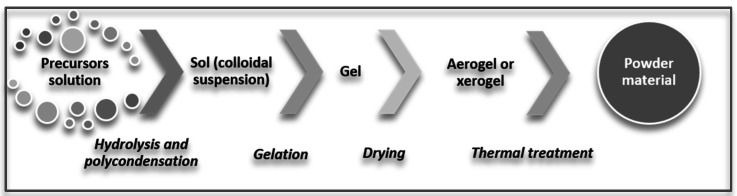
Synthesis scheme for sol-gel method.

**Figure 3 sensors-21-05758-f003:**
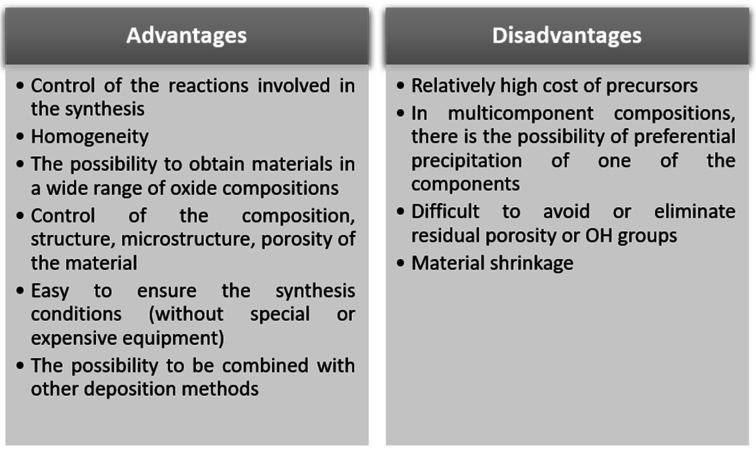
Advantages and disadvantages of sol-gel synthesis method.

**Figure 4 sensors-21-05758-f004:**
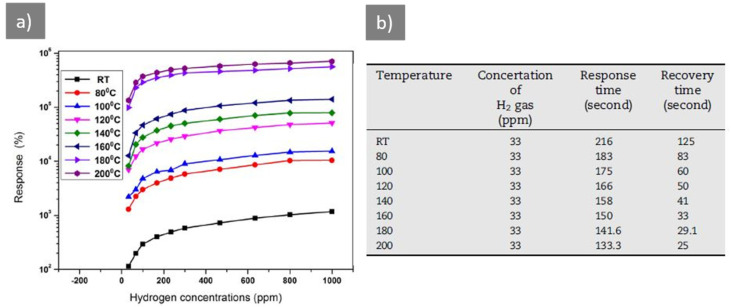
(**a**) Plot of the maximum responses measured after switching On/Off H_2_ gas every 5 min for 10 cycles for gas sensor upon exposure to different concentrations, from 33 to 1000 ppm, of H_2_ gas at different operating temperatures; (**b**) Response/Recovery time of SnO_2_-coated β-Ga_2_O_3_ NB s sensor for 33 ppm at different temperatures [70].

**Figure 5 sensors-21-05758-f005:**
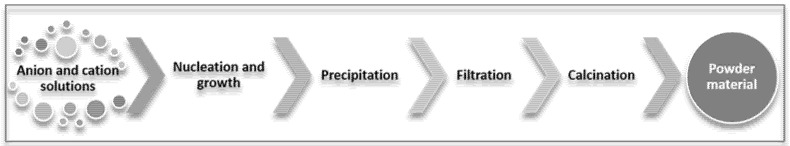
Synthesis scheme for the co-precipitation method.

**Figure 6 sensors-21-05758-f006:**
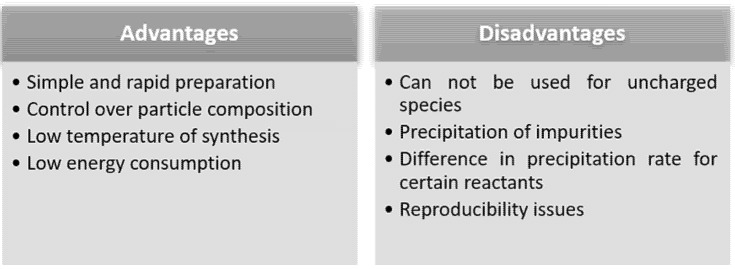
Advantages and disadvantages of co-precipitation method.

**Figure 7 sensors-21-05758-f007:**
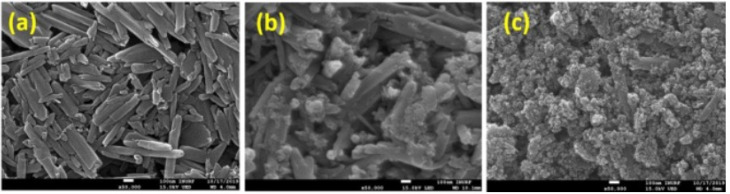
SEM images of (**a**) HNT s, (**b**) Fe_3_O_4_-HNTs and (**c**) Fe_3_O_4_-HNTs-Pd [95].

**Figure 8 sensors-21-05758-f008:**
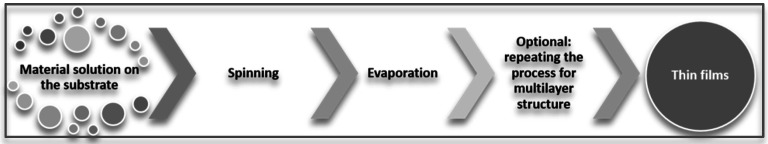
Synthesis scheme of spin-coating method.

**Figure 9 sensors-21-05758-f009:**
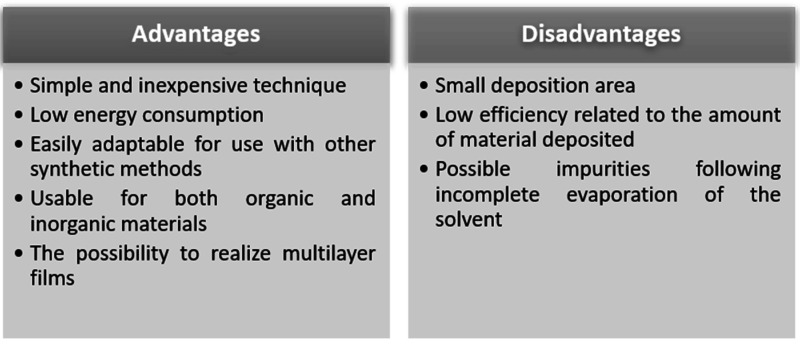
Advantages and disadvantages of the spin-coating deposition method.

**Figure 10 sensors-21-05758-f010:**
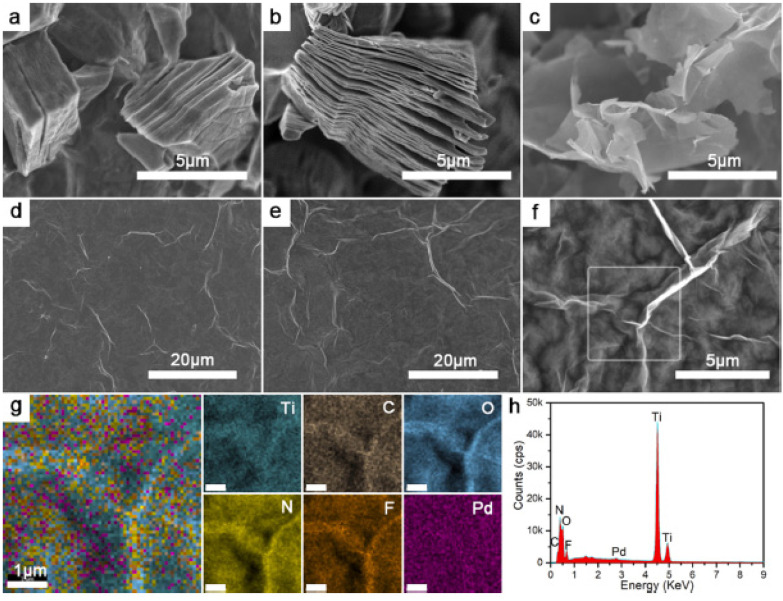
SEM and EDX images of the material surface morphology. (**a**) SEM image of bulk Ti_3_AlC_2_; (**b**) morphology of the commercial Ti_3_C_2_T_x_; (**c**) SEM image of d-Ti_3_C_2_T_x_ after intercalation; (**d**) morphology of/d-Ti_3_C_2_T_x_ film; (**e**–**h**) Pd (II)@alkyne-PVA/d-Ti_3_C_2_T_x_ film surfaces morphology, element mapping (1 µm scale bar), and EDX spectrum [109].

**Figure 11 sensors-21-05758-f011:**
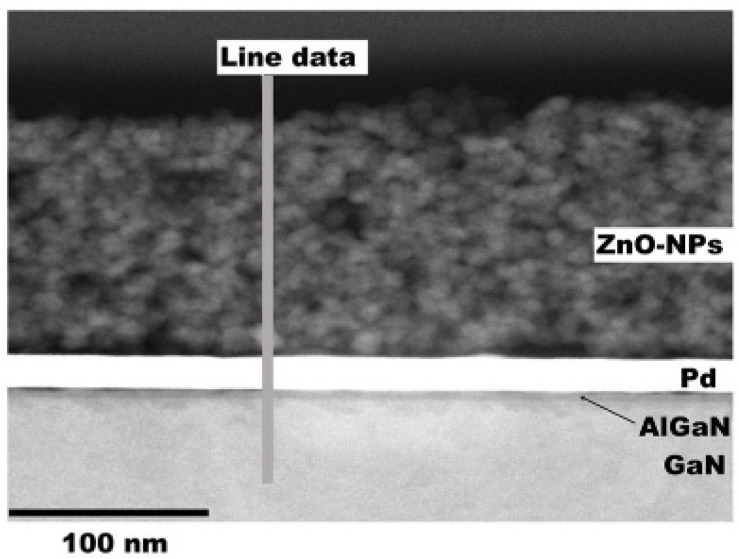
TEM analysis for ZnO-NPs/Pd dual catalyst layer of the AlGaN/GaN-on-Si hydrogen sensor [110].

**Figure 12 sensors-21-05758-f012:**
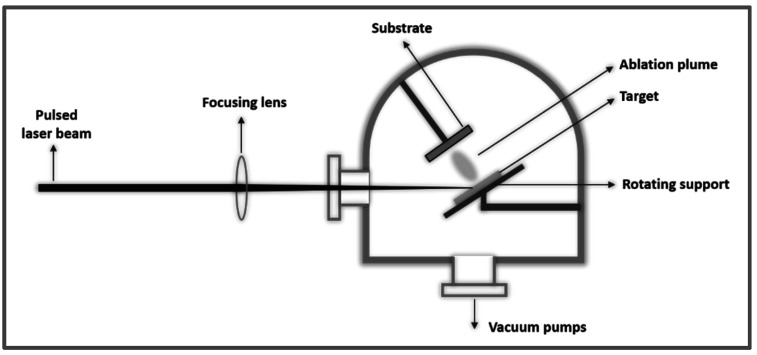
Diagram of PLD process.

**Figure 13 sensors-21-05758-f013:**
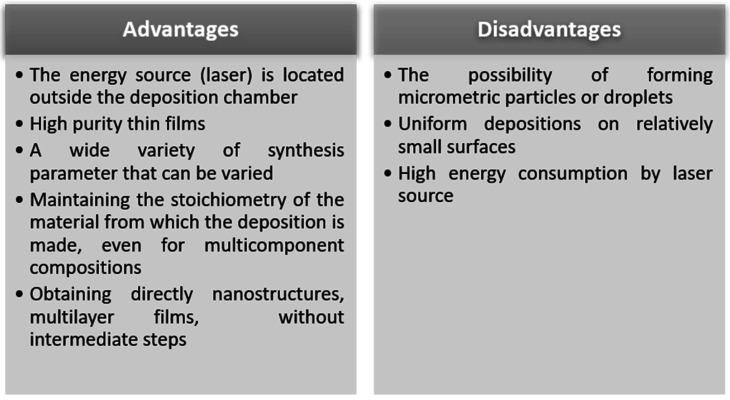
Advantages and disadvantages of PLD process.

**Figure 14 sensors-21-05758-f014:**
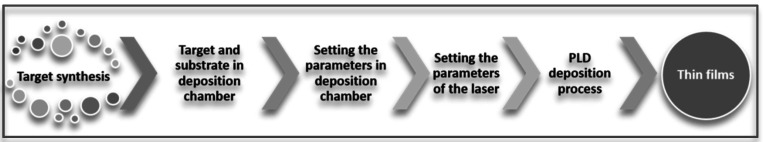
The scheme for the PLD deposition method.

**Figure 15 sensors-21-05758-f015:**
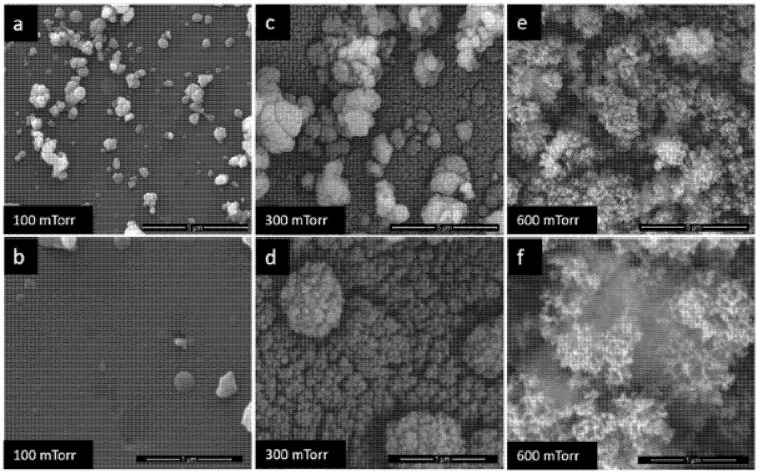
SEM images of TiO_2_ films at (**a**,**b**)—100 mTorr, (**c**,**d**)—300 mTorr, (**e**,**f**)—600 mTorr [122].

**Table 1 sensors-21-05758-t001:** Resistive sensors obtained by different synthesis methods.

Sensitive Material	Synthesis Method of the Sensitive Material	Morphology	Hydrogen Concentration	Response	Temperature	Reference
Pd	Evaporation	Nanoporous	2%	~0.037 (normalized resistance)	RT	[1]
p-TiO_2_ and Pd/p-TiO_2_	Sol-gel and dip coating	Nanoparticles with nanocracks	1% (in N_2_)	60.56 (%)	150 °C	[48]
Pd/SnO_2_/SiO_2_	RF Magnetron sputtering	Thin films	0.05%	611–1317%	RT	[50]
WO_3_/PdO	Precipitation	Nanorods	30,000 ppm	3.14 × 10^6^ (R_a_/R_g_)	150 °C	[52]
TiO_2_/Pd	DC Magnetron sputtering	Nanotubes	10 ppm	1.25 (ΔR/R_H2_)	180 °C	[51]
ZnO	Thermal oxidation	Nanosheets	10 ppm	1.089 (R_air_/R_H2_)	175 °C	[55]
Pd/SnO_2_	In situ self-assembling	Thin film	100 ppm	3 (V_g_/V_a_)	180 °C	[56]
0.6 wt% Pd/ZnO NFs	Electrospinning and electron beam irradiation	Nanofibres	0.1 ppm	74.7 (R_a_/R_g_)	350 °C	[53]
Pd/Mg	Pulsed Laser Deposition	Thin films	2 bar H_2_ gas atmospheres	37%	RT	[54]
Pd@ZnO-In_2_O_3_	Hydrothermal	Core-shell nanoparticles	100 ppm	42 (R_a_/R_g_)	300 °C	[28]
Al-Mg co-doped ZnO	Sol-gel in supercritical conditions	Nanoparticles	2000 ppm	70 (R_a_/R_g_)	250 °C	[2]

## Data Availability

Not applicable.

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
