# Peer review of "Synthesis Methods of Obtaining Materials for Hydrogen Sensors"

_sensors, 2021, doi:10.3390/s21175758_

Round 1

Reviewer 1 Report

The authors summarized the progress of synthesis and deposition methods of materials for hydrogen sensors including sol-gel, co-precipitation, spin-coating, and pulsed laser deposition (PLD). Overall, this review is informative, but I have some suggestions and comments in my mind at this moment:

  1. The title indicates synthesis and deposition methods. I wonder what’s the difference between synthesis and deposition. Is there any overlap? Based on my understanding, synthesis refers to create new things while deposition refers to change the status or morphology of a material. Actually, deposition can also “synthesize” new materials such as doping or oxidation.
  2. There are too many short paragraphs in the introduction part, chopping the information into too many small pieces. I suggest the authors re-organize the introduction part.
  3. The authors claimed the high coefficient of diffusion of hydrogen gas in air (0.6 cm2/s), what’s the influence of temperature?
  4. It is weird to shape the Advantages and Disadvantages into a figure in a scientific paper. People usually do that in a PowerPoint slide. In that regard, I would like to suggest removing Figure 3, Figure 6, Figure 9, and Figure 13 in the manuscript. Similarly, I do not think Figures 2, 5, 8, 14 are informative. Instead, giving some related examples that show the synthesis process will be convictive and deliver more ideas to the reader. Figure 16 is blurry. The authors should carefully think about how to re-design the figure set.
  5. In Table 1, how about the sensitivity of these resistive hydrogen sensors? How to define sensitivity?
  6. The alphabetic labels (a), (b), (c)… in the Figures are not consistent.
  7. I’m curious about why the authors want to show the Equations on Page 5. It’s very straightforward that higher spin speed will afford thinner films. So, my question is how these two Equations in the manuscript can be used to guide the material preparation except for spin speed.
  8. I would like to suggest deleting the first paragraph of the Conclusion section.
  9. There are some grammar issues and typos in the manuscript, such as:

In line 32, it should be other industries rather than another industry;

In line 51, the reference number should be incorporated by the Full stop;

In line 57, H2S should be H2S;

In line 61, it is a little tedious that says “Some of the types of sensors”;

……

To this end, I would like to suggest the author polishing the writing carefully.

Author Response

We thank you for the time and interest you have given to this work and for the desire to clarify certain important notions. We have tried to answer your questions as accurately and clearly as possible.

  1. The title indicates synthesis and deposition methods. I wonder what’s the difference between synthesis and deposition. Is there any overlap? Based on my understanding, synthesis refers to create new things while deposition refers to change the status or morphology of a material. Actually, deposition can also “synthesize” new materials such as doping or oxidation.

Answer: Thanks for the suggestion. The title has been changed. Initially we put synthesis and deposition, as in the article there are two methods of chemical synthesis and two methods of physical deposition, although those of deposition also presuppose synthesis.

  1. There are too many short paragraphs in the introduction part, chopping the information into too many small pieces. I suggest the authors re-organize the introduction part.

Answer:  The introductory part has been reorganized.

  1. The authors claimed the high coefficient of diffusion of hydrogen gas in air (6 cm2/s), what’s the influence of temperature?

Answer:  You're right, hydrogen gas diffusion is influenced by temperature. 0.6 cm2/s is the value of coefficient of diffusion at 0 ℃. As the temperature increases, the diffusion coefficient of hydrogen in the air increases, reaching up to 3.2 cm2/s at 400 ℃.

(Engineering ToolBox, (2018). Air - Diffusion Coefficients of Gases in Excess of Air. [online] Available at: https://www.engineeringtoolbox.com/air-diffusion-coefficient-gas-mixture-temperature-d_2010.html [Accessed 18.08.2021])

  1. It is weird to shape the Advantages and Disadvantages into a figure in a scientific paper. People usually do that in a PowerPoint slide. In that regard, I would like to suggest removing Figure 3, Figure 6, Figure 9, and Figure 13 in the manuscript. Similarly, I do not think Figures 2, 5, 8, 14 are informative. Instead, giving some related examples that show the synthesis process will be convictive and deliver more ideas to the reader. Figure 16 is blurry. The authors should carefully think about how to re-design the figure set.

Answer: Taking into account that this work is a review paper, the mentioned figures have the role of summarizing the information from the text for an easier reading and comparison of the information. Figures 2, 5, 8, 14 have the role of helping the reader to follow more accurately the synthesis steps specific to each method.

Figures 3, 6, 9 and 13 summarize the advantages and disadvantages of each method described. Also in figures 2, 5, 8 and 14 is schematically represented the synthesis / deposition process.

Figure 16 has been changed.

  1. In Table 1, how about the sensitivity of these resistive hydrogen sensors? How to define sensitivity?

Answer:  We did not give the sensitivity in Table 1, because each source in the table responds to the sensors at different hydrogen concentrations and at different temperatures. We define the sensitivity as the change in signal of a device per unit change in the parameter to which the device is sensitive.

  1. The alphabetic labels (a), (b), (c)… in the Figures are not consistent.

Answer:  The alphabetic labels (a), (b)…are not consistent because most of the figures are taken from other works, cited and we did not want to make changes to them.

  1. I’m curious about why the authors want to show the Equations on Page 5. It’s very straightforward that higher spin speed will afford thinner films. So, my question is how these two Equations in the manuscript can be used to guide the material preparation except for spin speed.

Answer: Indeed, it is very easy to see that as the rotational speed increases, the thickness of the film is smaller. But the properties of the deposited material or of the solvent used are also important parameters for the control of the thickness of the films deposited by spin coating. There are applications that require a fine control of the thickness of the films, and these equations are useful to achieve this.

  1. I would like to suggest deleting the first paragraph of the Conclusion section.

Answer: The first paragraph from the conclusion was deleted.

  1. There are some grammar issues and typos in the manuscript, such as:

In line 32, it should be other industries rather than another industry;

In line 51, the reference number should be incorporated by the Full stop;

In line 57, H2S should be H2S;

In line 61, it is a little tedious that says “Some of the types of sensors”

To this end, I would like to suggest the author polishing the writing carefully.

Answer:  The changes have been made, and the paper was revised.

Reviewer 2 Report

Revision of “Synthesis and deposition methods of materials for hydrogen sensors”

The manuscript under review devoted to summarize the latest studies on the development of new materials by means of simple and relatively handily methods, which encourages the development of hydrogen sensors. The authors considered the following methods: sol-gel, co-precipitation, spin coating and pulsed laser deposition (PLD). Providing of such investigations is very important both from an academic point of view (giving new knowledge about the nature of the objects under study) and economic (reducing the costs of industrial companies when obtaining new materials for various areas of the civil sector).

The development of hydrogen sensors has acquired a great interest for researchers, due to its introduction as fuel in vehicles, but also for safety in other fields such as chemical industry, metallurgy, pharmaceutics or power generation. The properties of the material of the sensitive area of the sensor are of great importance for establishing its performance. Besides the nature of the material, an important role for its final properties is played by the synthesis method used and the parameters used during the synthesis. Thus, the authors of the presented review have shown recent results in the field of hydrogen detection, obtained using four of the well known synthesis and deposition methods: sol-gel, co-precipitation, spin-coating and PLD.

In manuscript all necessary information is captured by 16 figures and 1 table . There are 123 references, all of them are adequate and are reflected in the text.

After getting acquainted with the presented manuscript, a few small questions remained:

  1. In a number of chemical formulas, numbers are not indicated as subscripts. For example lines 57, 524, 538, 546, 550, 559 etc.
  2. In the list of references, it is necessary to draw up references in a uniform way, namely, somewhere there is doi and somewhere not. Line 560, 576 etc.
  3. Line 618 need to split words “WO3nanorods”.
  4. Conclusion must be needs to be revised because at the moment it looks like an introduction, not a summary.

The obtained results are important both for understanding the physical processes that occur in real objects and for the development of new materials. The described manuscript is sufficient, comprehensive and it corresponds to the field of the Journal «Sensors». It may be accepted after minor revision.

Author Response

Thank you for the review. Regarding the suggestions made, we have done the following:

  1. In a number of chemical formulas, numbers are not indicated as subscripts. For example lines 57, 524, 538, 546, 550, 559 etc.

Answer: Changes have been made to the paper.

  1. In the list of references, it is necessary to draw up references in a uniform way, namely, somewhere there is doi and somewhere not. Line 560, 576 etc.

Answer:  We have done the modifications.

  1. Line 618 need to split words “WO3nanorods”.

Answer:  We have split the words.

  1. Conclusion must be needs to be revised because at the moment it looks like an introduction, not a summary.

Answer:  We revised the conclusions.

Reviewer 3 Report

This review paper describes ‘Synthesis and deposition methods of materials for hydrogen Sensors’. Authors are recommended to use these recent literatures on gas sensors for the detection of hydrogen and enhance the literature reviews in the introduction section have provided sufficient literature reviews, they are suggested to refer the following articles and improve the literature reviews on the gas sensors for the detection of hydrogen;---https://doi.org/10.1016/j.jpcs.2020.109864. 

---https://doi.org/10.1007/s10854-020-04387-3.  ---https://doi.org/10.1016/j.surfin.2021.101190  

---https://doi.org/10.1016/j.snb.2020.128330. ---https://doi.org/10.1016/j.jallcom.2020.154105 

Authors reported that sensors with very good results have been achieved by these methods, which gives an encouraging perspective for the use of these methods in obtaining commercial hydrogen sensors and their application in common areas for society. Additionally, authors reported various synthesis and deposition methods which have been used for the fabrication of gas sensors and sensing materials preparation including sol-gel, co-precipitation, spin-coating, hydrothermal, sputtering and pulsed laser deposition (PLD). These techniques have advantages over, authors are recommended to explain these advantages by using the following recent and relevant references;---https://doi.org/10.1002/pc.26034. ---https://doi.org/10.1039/D1TC01974K.   ---https://doi.org/10.1016/j.ceramint.2021.03.025.  ---DOI 10.24425/amm.2021.135877.---https://doi.org/10.1016/j.mssp.2020.105506.  ---https://doi.org/10.1016/j.apt.2020.10.015 

Manuscript has however some grammatical mistakes which need to be improved. As a whole, the topic and presented results are interesting. Manuscript contains novelty and it is well organized and authors systematically addressed the corresponding issues. I recommend publication of this article after major mandatory revisions and would like to see the revised version of paper before possible publication.

Author Response

We thank you for the time and interest you have given to this work.

We have corrected the grammatical errors, and we have added some of the references recommended by you.

Round 2

Reviewer 1 Report

Thanks for the revision. I think the authors have addressed all my concerns.

Reviewer 3 Report

However, authors incorporated only some of my comments, I recommend publication of this paper in the revised form.